# A Pose Awareness Solution for Estimating Pedestrian Walking Speed

**Guangyi Guo** [1] , **Ruizhi Chen** [1,2,*] , **Feng Ye** [1] , **Liang Chen** [1] , **Yuanjin Pan** [1] , **Mengyun Liu** [1] **and Zhipeng Cao** [1]

1 State Key Laboratory of Information Engineering in Surveying, Mapping and Remote Sensing (LIESMARS), Wuhan University, Wuhan 430079, China; guangyi.guo@whu.edu.cn (G.G.); yefeng92@whu.edu.cn (F.Y.); l.chen@whu.edu.cn (L.C.) pan_yuanjin@163.com (Y.P.); amylmy@whu.edu.cn (M.L.); godsay1983@whu.edu.cn (Z.C.)
2 Collaborative Innovation Center of Geospatial Technology, Wuhan University, Wuhan 430079, China
\* Correspondence: ruizhi.chen@whu.edu.cn; Tel.: +86-027-87731869 (ext. 801)

**Abstract:** Pedestrian walking speeds (PWS) can be used as a "body speedometer" to reveal health status information of pedestrians and positioning indoors with other locating methods. This paper proposes a pose awareness solution for estimating pedestrian walking speeds using the sensors built in smartphones. The smartphone usage pose is identified by using a machine learning approach based on data from multiple sensors. The data are then coupled tightly with an adaptive step detection solution to estimate the pedestrian walking speed. Field tests were carried out to verify the advantages of the proposed algorithms compared to existing solutions. The test results demonstrated that the features extracted from the data of the smartphone built-in sensors clearly reveal the characteristics of the pose pattern, with overall accuracy of 98.85% and a kappa statistic of 98.46%. The proposed walking speed estimation solution, running in real-time on a commercial smartphone, performed well, with a mean absolute error of 0.061 m/s, under a challenging walking process combining various usage poses including texting, calling, swinging, and in-pocket modes.

**Keywords:** smartphone; multi-sensors; posture context; walking speed estimation

## 1. Introduction

Due to the development of microelectronics technology, modern smartphones are equipped with a rich set of sensors and have been explored as a ubiquitous computing platform. With this platform, particular attention has been focused on the estimation of the walking speed of pedestrians, which, as a "body speedometer," can reveal pedestrians health status information, such as joint strength [1] and lifestyle [2], and predict future health [3]. A precise walking speed is necessary for the use of many location-based services as well, especially in the indoor environment where most global navigation satellite system (GNSS) signals are blocked.

The current pedestrian walking speed estimation methods can be divided into two categories: radio frequency (RF)-based methods and sensor-based methods. In RF-based walking speed estimation and activity recognition, there have been several interesting studies in recent years [4–9]. For example, Shi [7] used the fluctuation in ambient FM radio signals to infer pedestrian attention levels by interpreting changes in their walking speed and direction. Sigg [8] considered the detection of activities from noncooperating individuals with features obtained on a radio frequency channel and used the WiFi received signal strength information (RSSI) on a smartphone to estimate walking speed [9]. Generally, these types of approaches exploit both the time and frequency domains of the statistical features (e.g., mean, variance, kurtosis, and skewness), and use a machine learning method

(i.e., k-nearest neighbor decision tree) to classify the walking speed. That is, the speed estimation problem is transformed into a classification problem. Therefore, only a qualitative walking speed can be determined.

The sensor-based algorithms use smartphone inertial sensors (e.g., accelerometer, gyroscope, and magnetometer) to estimate the pedestrian walking speed and can be further divided into two subcategories, i.e., machine learning (ML)-based methods [10–14] and speed model-based methods [15–30]. The ML-based methods have the potential to exploit associative information within the data beyond an explicit model chosen by the system designer [10]. In principle, rather than using a certain physical model to estimate the walking speed, the ML-based approach expresses the complex relationship between the measurements from the inertial sensors and the walking speed by training a black-box model [11]. Several research studies have focused on use of a regression model and ML techniques to improve the precision of the walking speed estimation. Vathsangam [12] proposed a nonlinear and nonparametric regression framework to estimate the walking speed from an accelerometer fixed on a subject's hip. Park [10] estimated the speed from the energy of acceleration magnitude by applying regularized kernel methods on collected accelerometer data to achieve a higher accuracy of walking speed estimation. Yeoh et al. [13] estimated the speed by using a third-order polynomial model that fits the mean value of the average net acceleration (ANA). However, the automatic selection and extraction of notable features remain a challenge. Consequently, a deep convolutional neural network (DCNN) is applied to automatically identify and extract the most effective features from the accelerometer and gyroscope data of the smartphone and to train the network model for speed estimation [14]. Similarly, the deep learning method generally requires a large amount of labeled training data. As the neural network grows larger and deeper, it becomes more difficult to train the network to perform a task. The other group of sensor-based method is model-based approaches, in which the smartphone can be regarded as a pedometer that uses sensor measurements to detect step events. Cox [15] proposed a simple solution to estimate walking speed based on the integration of the acceleration. Cho [16] proposed the opportunistic calibration of the inertial sensor-based speed estimation using the GPS of a smartphone when the user is walking outdoors. Other pedestrian navigation systems have also considered methods of speed estimation, typically using inertial and magnetic sensors along with heuristic- or rule-based speed estimation [10,17,18]. Masaru [19] generated magnetic signatures and obtained a walking speed from walking distance and walking time by using dynamic time warping (DTW). It should be noted that the accuracy of the step length model has a large effect on the final precision of the model-based approaches. Several methods have been proposed to estimate the step length, including human gait-based [20,21], step frequency-based [22,23], and step counting-based methods [24–26]. However, these pedestrian walking speed estimation methods suffer from various limitations such as unsuitability for smartphone-based applications [27], a lack of consideration of different pose context [28,29], user-dependency [22,23], and reliance on spatial constraints [24,25,30].

In this work, we aim to develop an adaptive pedestrian walking speed estimation solution that provides pose context awareness and is therefore capable of achieving high accuracy using a normal smartphone. This approach is tightly coupled with real-time pose identification and pedestrian walking information using an adaptive step detection strategy. The multi-sensor data is collected from sixty-one male and thirty-eight female subjects and labeled with the pose type, and these data are used to evaluate the extracted features and train the classifier. To assess the performance of the proposed pedestrian walking speed solution, various field tests are carried out in an indoor environment, and the effectiveness of the solution is verified by comparison with the results from a Leica total station.

The rest of the paper is organized as follows: the system architecture and methodology of the proposed system are demonstrated in Section 2. In Section 3, the experimental platform is described in detail, and numerical results and a performance comparison are presented. Sections 3.3 and 3.3 provide a discussion and conclusion of the whole work, respectively, and give suggestions for future research.

## 2. Materials and Methods

The architecture of the proposed pedestrian walking speed estimation solution on a smartphone is illustrated in Figure 1. In addition to the use of motion sensors (i.e., tri-accelerometer and tri-gyroscope), the estimation process also takes into consideration information obtained from ambient light and proximity sensors. All sensor data are gathered by the developed customized application developed and the Android phone is run with a sampling frequency of 100 Hz. In Figure 1, after receiving the data, the usage pose context awareness module, step detection module and pedestrian walk detection module start to operate. If the walking detection module decides that the pedestrian is in static mode, the zero speed is updated immediately. Alternatively, if the pedestrian is in walking mode, the pose mode is detected by using multi-sensor data and the ML method every 0.6 s. Next, step detection and step length estimation are executed to calculate the step frequency and current step length both with the aid of context information. Finally, the pedestrian walking speed is derived from all of these estimated results. The following sections describe each part of the pedestrian walking speed estimation system as well as its features and advantages.

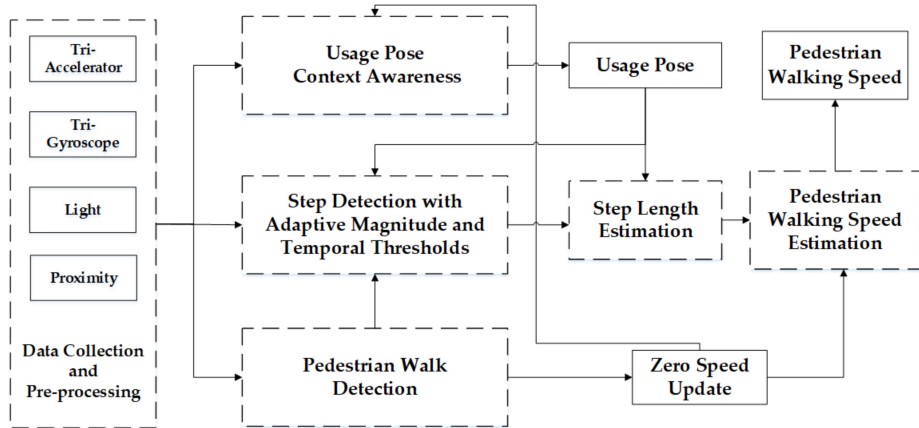

**Figure 1.** Scheme of speed estimation.

*2.1. Usage Pose Context Awareness Based on Multi-Sensor Data*

2.1.1. Usage Pose Context Definition

The usage pose context is defined as a series of motion patterns [31] that can be detected with a consumer-grade smartphone. However, smartphones experience a large variety of unrestrained and personal motions, which generate different patterns in sensors. In this section, the daily use modes are divided into two major categories with four subcategories to cope with the complexity of the modes. Based on observations of daily smartphone usage habits and contrasted with previous studies [26,32,33], as shown in Figure 2, four basic usage pose contexts of smartphone covered in this study are considered.

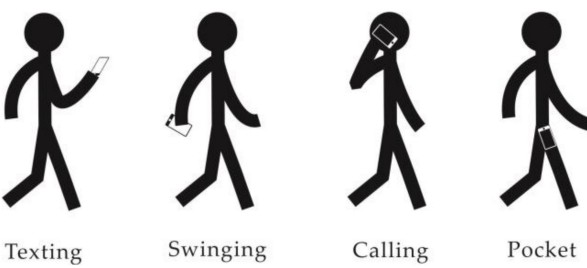

Texting          Swinging          Calling          Pocket

**Figure 2.** Four basic pose contexts.

**Relatively steady state:** This category includes all situations in which the motion state of a mobile device is relatively stable and there is no dramatic relative movement between the mobile device and the user's body. This includes the following cases:

- Hand texting: This case is the smartphone use case. To perform operations such as text messaging or reading the news, the user's eyes, hand, and screen should remain relatively stationary.
- Hand calling: In hand calling, the user makes a phone call while walking or remaining stationary. Intuitively, user's ear, hand, and phone should remain relatively stationary.
- In pocket: The user carries the mobile device in a pocket.

**Relatively dynamic state:** This class refers to the hand-swinging case in which the user is walking while holding the mobile device in his/her swinging hand. Relatively cyclical swinging occurs between the smartphone and the user's body while walking

### 2.1.2. Feature Extraction

Feature selection and extraction play vital roles in processing of the pattern recognition and have a significant effect on the performance and the final precision. In this section, statistics for multi-sensors time-series data are collected to detect the posture context, including the motion context and usage environment context. The readings from accelerometer and gyroscope can reflect dynamic changes in the usage mode, which are used to calculate the pitch and roll of a smartphone. The statistics for pitch and roll in the time series can be used to mine the pose pattern. Statistics for light and proximity sensors data are used to perceive the smartphone usage environment context, such as bright or dark locations, and the proximity to the body. For example, if we know that the light sensor value and proximity sensor value are both quite low, we can speculate that the mobile device is near the ear or in a pocket. Therefore, the mobile device usage environment should be detected simultaneously.

As shown in Figure 3, the coordinate system of the sensors in a smartphone is defined with the screen of the phone and its default orientation. Pitch and roll are the rotation around the x-axis and the rotation around the z-axis, respectively.

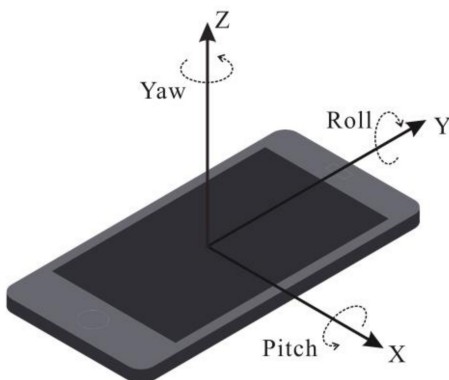

**Figure 3.** Smartphone reference frame.

When the pose mode changes, for example, texting mode transit into swinging mode, the pitch and roll change synchronously and sensitively and demonstrate different patterns. As shown in Figure 4, in a relatively steady state (hand texting, hand calling, in the pocket), the mobile device remains relatively static with the body. Therefore, the pitch and roll angles are relatively significant in those processes, whereas in a relatively dynamic state (hand swinging), pitch and roll angle change only periodically.

The statistics of the pitch, roll, and the readings from the light and proximity sensors in the sliding window are studied in this research. The details are listed in Table 1. The size of the sliding windows $N$ is another critical factor that affects the performance. The size of the sliding window should be long

enough to be able to observe transitions of a sudden motion mode but short enough to maintain the efficiency of the algorithm. The windows size $N$ is selected as 0.6 s with a 100 Hz sampling frequency in this paper.

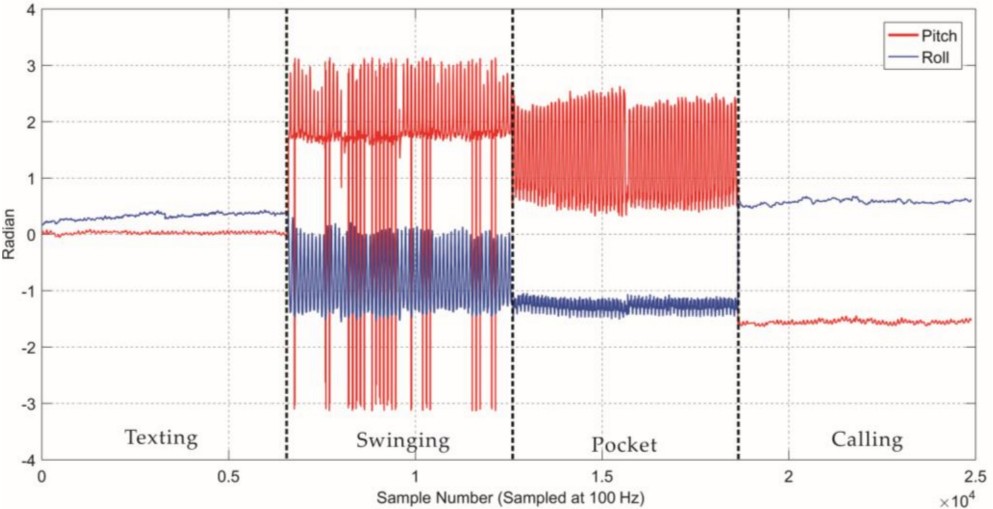

**Figure 4.** Example of the pitch and roll data in different poses.

**Table 1.** Descriptions of the features.

| Features | Sensor | Description |
|---|---|---|
| $P_{a_\mu}$ | Accelerometer/Gyroscope | Mean of the pitch angle over the moving window |
| $P_{a_{med}}$ | Accelerometer/Gyroscope | Median of the pitch angle over the moving window |
| $P_{a_{\max}}$ | Accelerometer/Gyroscope | Maximum value of the pitch angle over the moving window |
| $P_{a_{\min}}$ | Accelerometer/Gyroscope | Minimum value of the pitch angle over the moving window |
| $P_{a_{rms}}$ | Accelerometer/Gyroscope | Root mean square of the pitch angle over the moving window |
| $P_{a_{range}}$ | Accelerometer/Gyroscope | Range of the pitch angle over the moving window |
| $R_{a_\mu}$ | Accelerometer/Gyroscope | Mean of the roll angle over the moving window |
| $R_{a_{med}}$ | Accelerometer/Gyroscope | Median of the roll angle over the moving window |
| $R_{a_{\max}}$ | Accelerometer/Gyroscope | Maximum value of the roll angle over the moving window |
| $R_{a_{\min}}$ | Accelerometer/Gyroscope | Minimum value of the roll angle over the moving window |
| $R_{a_{rms}}$ | Accelerometer/Gyroscope | Root mean square of the roll angle over the moving window |
| $R_{a_{range}}$ | Accelerometer/Gyroscope | Range of the roll angle over the moving window |
| $L_\mu$ | Light | Mean of the light value over the moving window |
| $L_{med}$ | Light | Median of the light value over the moving window |
| $L_{rms}$ | Light | Root mean square of the light value over the moving window |
| $P_\mu$ | Proximity | Mean of the proximity value over the moving window |
| $P_{med}$ | Proximity | Median of the proximity value over the moving window |
| $P_{rms}$ | Proximity | Root mean square of the proximity value over the moving window |

### 2.1.3. Classification

Various ML algorithms, such as naïve Bayes [34], k-nearest neighbor [35,36], decision tree (DT) [36], neural network [37], support vector machines [38,39], and random forest (RF) [40], etc. are used for the purpose of posture context recognition. Meanwhile the merits and drawbacks of those ML methods are compared and analyzed in many studies [41,42]. These studies demonstrated that RF offers a number of advantages, such as a straightforward learning process, ease of parallelization, a shorter training time, and a higher travel prediction accuracy. Consequently, only the RF methodology is used to solve this real-time classification problem in this paper.

Finding the mean noise is the core step of this methodology. RF is an ensemble of binary decision trees, which grow to their maximum depth and reduce the relevance of the individual decision trees by using randomization. This randomness introduces robustness against noise to the algorithm. According to the strong law of large numbers, as the number of decision trees in a random forest increases, the generalization error converges to a limit, and thus the overfitting can be

effectively avoided. The generalization error is dependent on the strength of the individual trees and their correlation.

### 2.2. Pedestrian Walking Speed Estimation

#### 2.2.1. Walk Detection

Pedestrian walk detection (PWD) is a highly critical step in estimating the walking speed and determining the geospatial location. PWD can effectively avoid unnecessary and expensive computing during the non-motion and promotes the accuracy of the speed estimation. The geospatial location of the user does not significantly change in static state, such as standing still, texting, reading news, answering a phone call, or turning around to find a location. However, with the unconstrained and personal variation in the use of smartphones, there is no absolute static state or zero speed during the speed estimation task or pedestrian navigation.

In this paper, two thresholds were set to detect the pedestrian walking state by using the accelerometer and gyroscope readings. Due to the units of angular rate and acceleration being different, the norm of the output vector from tri-accelerometer and tri-gyroscope was used to detect the pedestrian walk. The dynamic response of the accelerometer is slow, and its range of measurement is restricted. The gyroscope offers advances in excellent dynamic performance and sensitivity; however, the data suffer from the changes of the temperature and unstable torques and thus produce drift errors.

$$\delta_{\min} < \frac{1}{N} \sum_{k \in \Omega_n} \left( \left\| y_k^a - \overline{y_k^a} \right\|^2 + \alpha \left\| y_k^g \right\|^2 \right) < \delta_{\max} \tag{1}$$

where $N$ is the size of the sliding windows for PWD which was selected as 0.6 s, $y_k^a$, and $y_k^g$ represent the readings of the tri-accelerometer and tri-gyroscope, respectively, and $\overline{y_k^a}$ represents the mean value of the accelerometer output vector in the sliding window. $\alpha = 9.8$ is a scale coefficient of gyroscope output vector. $\delta_{\min}$ and $\delta_{\max}$ represent the minimum and maximum PWD thresholds, respectively.

#### 2.2.2. Preprocessing

In general, the output of the tri-accelerometer and tri-gyroscope might appear in the form of harmonic oscillation waveforms caused by walking behaviors [32]. Using the repetitiveness and periodicity of the pedestrian's walking, the number of steps that a pedestrian has traveled can be computed.

Recently, some algorithms based on accelerometers data processing have been developed for step detection. In those approaches, the magnitude of the three axes $a_{mag,k}$ are used to analyze the step, and $a_{mag,k}$ can be expressed as:

$$a_{mag,k} = \sqrt{a_{x,k}^2 + a_{y,k}^2 + a_{z,k}^2} \tag{2}$$

However, the use of the magnitude to do the step detection task neglects the information of pose context implicit in the attitude of smartphone.

In this paper, the pedestrian steps were detected with knowledge of the different posture modes and its own corresponding sensor data. When a user walks with the smartphone in texting mode, calling mode, or pocket mode, the accelerometer signal can clearly present a cyclic pattern. When the user walks with the smartphone in swing mode, there is synchronization between the arm and foot motion, as has been shown by biomechanical studies. This synchronization relationship between the swinging arm and the reaction moment about the vertical axis of the foot is explained in the context of the dynamics of a multi-body articulated system [43]. Specifically, as a pedestrian walk, the positive torque produced by the arm swing makes the foot move forward. Therefore, a sinusoidal pattern in the gyroscope reading is used to detect the walking step.

As shown in Figure 5, for texting mode, the *y*-axis and *z*-axis acceleration signals show obvious periodicity, and the feature signal $\gamma_{T,k}$ can be expressed as:

$$\gamma_{T,k} = \sqrt{a_{y,k}^2 + a_{z,k}^2} \tag{3}$$

where $a_{y,k}$ and $a_{z,k}$ denote the readings of accelerometer on the *y*- and *z*-axes.

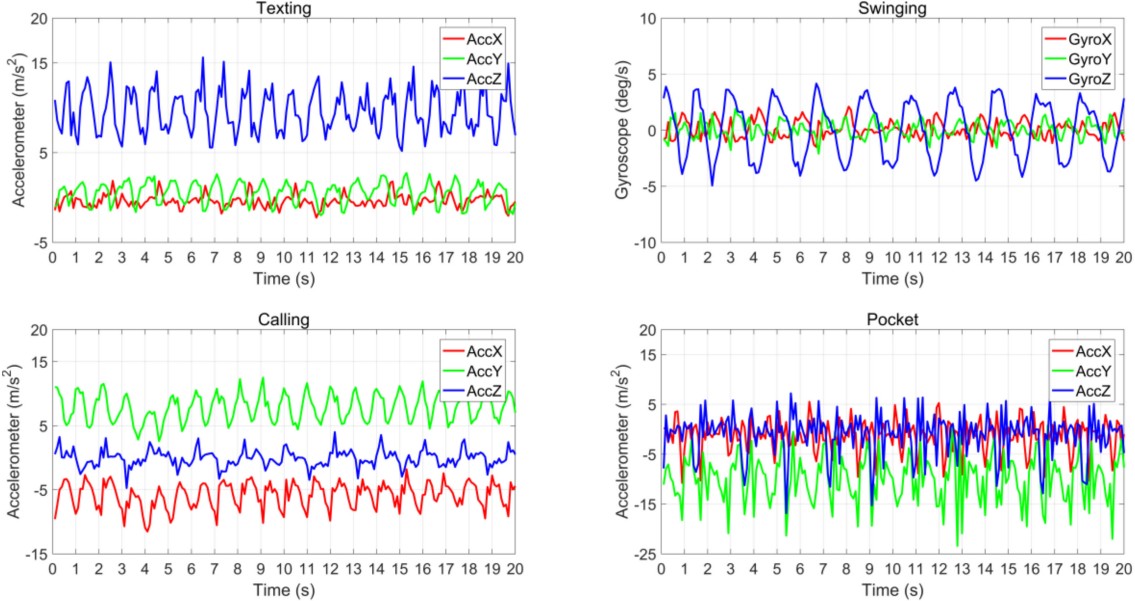

**Figure 5.** Internal sensors data in different poses.

For calling mode, the *x*-axis and *y*-axis acceleration signal shows obvious periodicity, and the feature signal $\gamma_{C,k}$ can then be expressed as:

$$\gamma_{C,k} = \sqrt{a_{x,k}^2 + a_{y,k}^2} \tag{4}$$

where $a_{x,k}$ $a_{y,k}$ denotes the readings of accelerometer on the *x*- and *y*-axes.

For pocket mode, the magnitude of the tri-axis accelerometer is used to detect the step. The feature signal $\gamma_{P,k}$ can then be expressed as:

$$\gamma_{P,k} = \sqrt{a_{x,k}^2 + a_{y,k}^2 + a_{z,k}^2} \tag{5}$$

where $a_{x,k}$ $a_{y,k}$ and $a_{z,k}$ denotes the readings of accelerometer on the *x*-, *y*- and *z*-axes.

For swing mode, the *z*-axis gyroscope signal shows an obvious periodicity, and the feature signal $\gamma_{S,k}$ can be expressed as:

$$\gamma_{S,k} = |g_{z,k}| \tag{6}$$

where $g_{z,k}$ represents the readings of gyroscope on the *z*-axis.

To minimize the impact of the mobile device shaking and sensor drift, and to improve the robustness of the step detection algorithm, a 10th order Butterworth filter [44] with a 3-Hz cut-off frequency was used for preprocessing of the time-series sensors feature signal. The purpose of the above pre-processing phase was to extract the signal's fundamental frequency that is induced by step events only and therefore only from an undistorted signal [45]. In this manner, the interference from high-frequency noise and unstable output sensors data can be reduced. In Figure 6, the raw data and the output after filtering are compared.

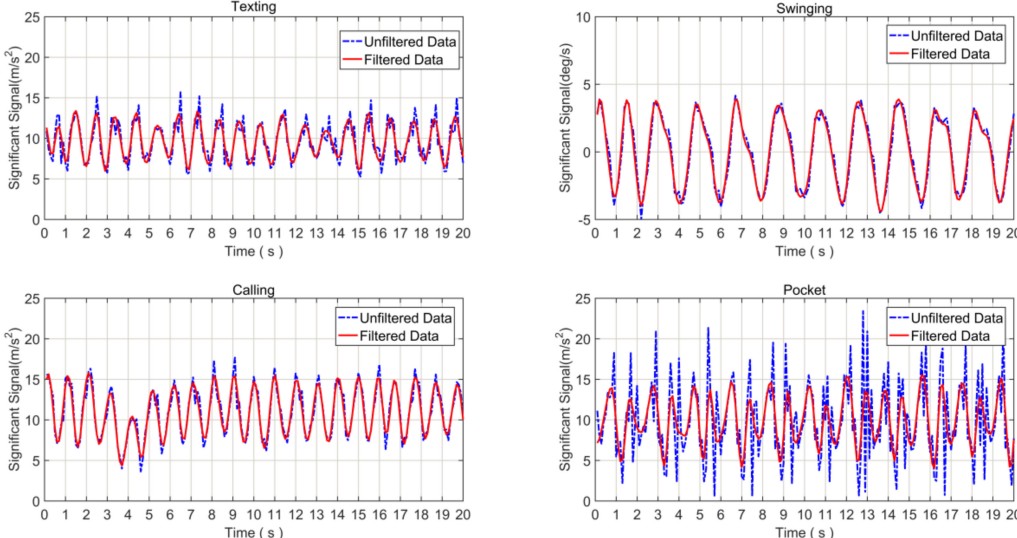

**Figure 6.** Comparison of the raw data and the output after filtering.

### 2.2.3. Step Detection with Adaptive Magnitude and Temporal Thresholds

Pedestrian walking is a continuously changing process that has the three characteristics of periodicity, similarity, and continuity. With the consideration of those characteristics and calculation time, two adaptive thresholds were established from two different aspects. For the vertical aspect, magnitude thresholds were used to detect the step point using peak detection approach. The magnitude of a step point should be the local maximum and larger than the adaptive threshold. The threshold was adapted dynamically based on the magnitudes of previous steps. The threshold consists of the average and standard deviation of the magnitude of the acceleration in a fixed window. However, increasing the window size might degrade the step detection accuracy during the transition of a step mode or device pose, because the threshold calculated from a larger window might be unable to effectively handle the variation in the recent statistics. From the horizontal aspect, temporal thresholds were used to constrain the step point in the step frequency dimension. Walking is an ongoing and relatively stable process, and therefore, the time interval between two steps and the time change of two steps should be within the range of normal human levels. On the basis of this analysis, the following criteria are defined in this paper:

**Criteria 1.** The feature signal of the candidate step point should be the local maximum:

$$\left|\gamma_{mag,k}\right| > \max\left(\left|\gamma_{mag,k-1}\right|, \left|\gamma_{mag,k+1}\right|\right) \tag{7}$$

**Criteria 2.** The feature signal of the candidate step point should exceed the adaptive threshold according to the current motion mode:

$$\left|\gamma_{mag,k}\right| > \mu_{\gamma_{mag,win}} - \omega_M \frac{\sigma_{\gamma_{mag,win}}}{\alpha} \tag{8}$$

where $\mu_\gamma$ and $\sigma_\gamma$ represent the mean and standard deviation of the magnitude in a fixed window, respectively, and $\alpha$ and $\omega_M$ represent two magnitude scale constant based on the current pose context.

**Criteria 3.** The walking step frequency should be within the range of the frequency threshold from $v_{\min}$ to $v_{\max}$, which is the range of the normal level:

$$v_{\min} < v_{step} < v_{\max} \tag{9}$$

The proposed adaptive step detection is outlined in the pseudo code in Table 2 where $y_k^a$ and $y_k^g$ are the samples of tri-accelerometer and tri-gyroscope vector at sample time $k$, respectively, and $T_{win}$

is calculated according to Equation (1) for PWD. $M_c$ is the current pose context. $\mu_\gamma$ and $\sigma_\gamma$ represent the mean and standard deviation of the magnitude in a step fixed window, and $\alpha$, $\omega_M$, $v_{\min}$, and $v_{\max}$ represent the adaptive constants based on $M_c$.

**Table 2.** Pseudocode of Step Detection Algorithm.

---

**Input:** $y^a_{k-1}$, $y^a_k$, $y^a_{k+1}$, $y^g_{k-1}$, $y^g_k$, $y^g_{k+1}$, $M_c$
**Output: Step point** $S_{step}$
**Begin**: **calculate** $T_{win}$ according to Equation (1).
　　**if** $T_{win} < \delta_{\min}$ or $T_{win} > \delta_{\max}$
　　　**return false**
　　**else**
　　　**calculate** $\gamma_k$ according to $M_c$ and Equations (3)–(6).
　　**if** $|\gamma_k| > \max\left(|\gamma_{k-1}|, |\gamma_{k+1}|, \mu_\gamma - \omega_M \frac{\sigma_\gamma}{\alpha}\right)$
　　　　**if** $v_{step} \in \{v_{\min}, v_{\max}\}$
　　　　　**update** $\mu_\gamma$, $\sigma_\gamma$
　　　　　**return** $S_{step}$
　　　　**end if**
　　　**end if**
　　**end if**

---

### 2.2.4. Step Length Estimation

Many algorithms have been proposed to estimate the step length, including human gait-based, step frequency-based, and step counting-based methods.

Pratama [46] estimates the step length based on a static model that considers a constant relative to height and sex, where $H$ represents the height, and $k$ is equals to 0.415 for male and 0.413 for female subjects.

$$SL = k \cdot H \tag{10}$$

The approach of Weinberg [28] assumes that the gait impacts the vertical acceleration and uses the difference between the maximum and minimum values of the vertical acceleration in each step to estimate the step length. The model formula is:

$$SL = k \cdot \sqrt[4]{acc_{\max} - acc_{\min}} \tag{11}$$

where $acc_{\max}$ and $acc_{\min}$ represents the maximum and minimum vertical acceleration values measured in a single stride, respectively, and $k$ is a constant model parameter.

The model proposed by Tian [47] estimates the step length based on the step frequency, height, and sex of the subjects as:

$$SL = k \cdot H \cdot \sqrt{SF} \tag{12}$$

where $H$ and $SF$ represents the height of the subject and step frequency, and $k$ is a model parameter and that is tuned to 0.3139 for male and 0.2975 for female subjects.

Kim [29] develops an empirical method based on the dependence of the average acceleration on the step length during walking. The step length is calculated using this method as:

$$SL = k \cdot \sqrt[3]{\frac{\sum\limits_{k=1}^{N} |acc_k|}{N}} \tag{13}$$

where $acc_k$ is the acceleration measured on a sample in a single step and $N$ is the number of samples corresponding to each step.

Compared with the above step length model, in this paper, the empirical and linear model in Reference [22] is used to estimate the step length, representing the relationship among the pedestrian's

height, step frequency, and step length. The equation used to estimate the step length is written as follows:

$$SL = \left(0.7 + a(H - 1.75) + \frac{b(SF - 1.79)H}{1.75}\right)c \tag{14}$$

where $SL$ and $SF$ represent the step length and step frequency, respectively, $H$ is the height of the pedestrian which is manually inserted in this step model, and $a$, $b$, and $c$ are model parameters for each person and can be calibrated by pre-training.

## 3. Results

### 3.1. Experimental Setup

Two experiments were carried out in the indoor environment to test the proposed algorithms. In the first experiment, the performance of the usage pose context awareness algorithm was evaluated based on field tests carried out in the lobby of the Library of Wuhan University. Sixty-three male subjects and thirty-eight female subjects took part in this experiment. As shown in Figure 7, the subject heights varied from 155–192 cm, and their ages ranged from 17 to 53. Because of the subjects' different patterns in smartphone use, we subdivided the calling, swinging, and pocket modes into left-hand use and right-hand use. The percentages of swinging, pocket, and calling modes with the use of the left hand were 6.93%, 21.78%, and 9.9%, and the percentages of the three modes with the use of the right hand were 93.07%, 78.22%, and 90.1%, respectively. In the training process, sixty percent of the subjects' recorded multi-sensors data that were used to train the classifier, and in the testing part, the other forty percent of the subjects' data were used to test the performance of the classifier. Four smartphones, including Huawei Mate9, Huawei P9, Huawei P9 Plus, and Huawei Honor 8, were used in the experiment, and all with Android platforms. The sensor data, including the accelerometer, gyroscope, light sensor, and proximity sensor data, were collected and labeled by an android application in real time, and the sampling rate of the sensors was set to 100 Hz.

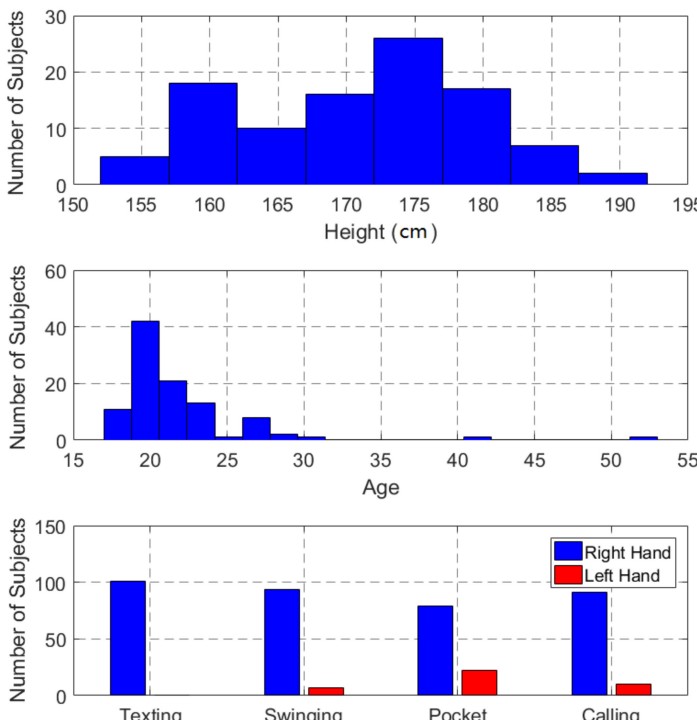

**Figure 7.** Statistics on subjects participating in our data collection: The height and age distributions of the subjects and the left-hand use and right-hand use ratio of subjects for different poses.

The second experiment was an evaluation of the walking speed estimation algorithm. In total, four smartphones (the same models as in the first test) were used in the experiment. Six men and six women participated in this experiment, with heights from 158–183 cm and of ages 22–53. The sensors data, including the accelerometer, gyroscope, light sensor, and proximity sensor data, were collected using an Android application in real time, and the sampling rate of sensors was set of at 100 Hz. The ground truth of the walking speed was measured using the Leica Nova TS60 total station, which can track a 360-degree prism automatically and supply one observation every 0.15 s with 3-mm precision. As shown in Figure 8, a participant carried the 360-degree prism on his/her back and walked 80 m with four different postures. The ground truth of the step count was read from videos taken during the entire experiment.

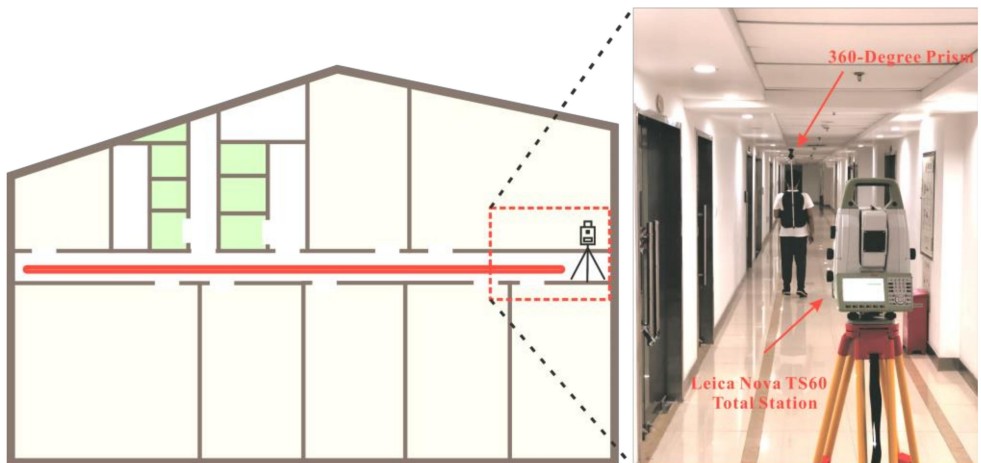

**Figure 8.** Experimental site. A Leica Nova TS60 total station was placed at the end of the corridor and automatically tracked a 360-degree prism. Participants carried the 360-degree prism on their back and held the smartphone in their hand.

### 3.2. Perfomance Evaluation of Usage Pose Awareness

Figure 9 provides the precision and recall rate of the proposed pose context awareness. The average precision rate of the proposed algorithm was 98.833% and the average recall rate was 98.828%. Table 3 shows a confusion matrix of the classification results. For the proposed algorithm, an overall accuracy (OA) of 98.85% and a kappa statistic (KS) of 98.46% were obtained. User's accuracy (UA) and producer's accuracy (PA) were above 97.88% in all the classes, and the commission error (CE) and omission error (OE) were within 2.12% in all the classes.

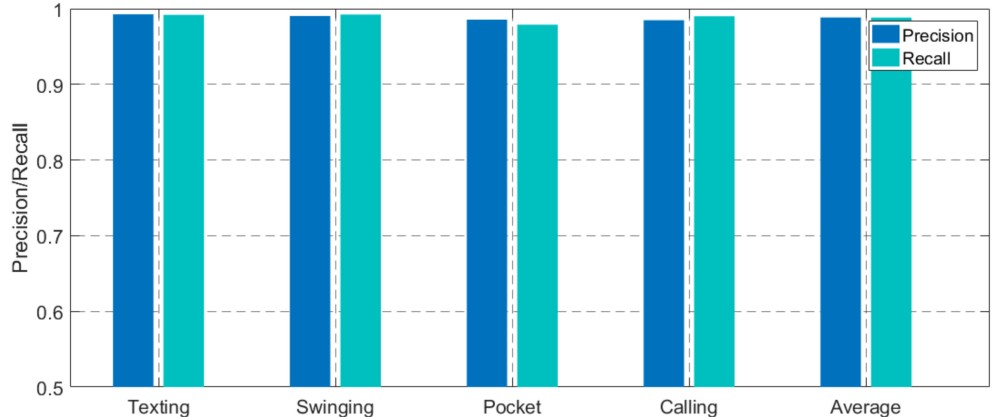

**Figure 9.** Average accuracy of the proposed posture context awareness algorithm. The rightmost two bars are the average precision and recall rate of the four different poses.

**Table 3.** Confusion matrix of pose recognition with seven categories.

| | | Predicted Class | | | | | | |
|---|---|---|---|---|---|---|---|---|
| | Class | Texting | Swinging | Pocket | Calling | Total | PA | OE |
| **Actual class** | **Texting** | 1219 | 2 | 8 | 0 | 1229 | 99.19% | 0.81% |
| | **Swinging** | 2 | 1031 | 5 | 1 | 1039 | 99.23% | 0.77% |
| | **Pocket** | 3 | 3 | 1018 | 16 | 1040 | 97.88% | 2.12% |
| | **Calling** | 4 | 5 | 2 | 1101 | 1112 | 99.01% | 0.99% |
| | **Total** | 1228 | 1041 | 1033 | 1118 | 4420 | | |
| | **UA** | 99.27% | 99.04% | 98.55% | 98.48% | | **OA** | 98.85% |
| | **CE** | 0.73% | 0.96% | 1.45% | 1.52% | | **KS** | 98.46% |

We also divided the swinging, pocket, and calling modes into two subclasses of left-handed and right-handed use, and tested the performance of the extracted feature and classifier further. As shown in Table 4, the overall accuracy (OA) and kappa statistic (KS) of the seven categories were 98.85% and 98.46%, respectively, which were both lower than the previous classification results, but the proposed method still delivered a high level of precision. From the result, it is clear that the recognition accuracy of left-hand use was lower than that of right-hand use. This was due to the following two reasons. First, as shown in Figure 7, fewer subjects were tested with their left hand than with their right hand, which resulted in insufficient samples and imbalanced training data. Second, the extraction features for the patterns of left-handed and right-handed use were highly similar, and misclassification within the major categories could lead to a remarkable decrease in the precision of the posture context awareness. For example, the users' accuracy of all pocket mode was 98.55%, but the users' accuracy of left-handed pocket mode and right-handed pocket mode were 94.55% and 96.43%, respectively.

**Table 4.** Confusion matrix of pose recognition with seven categories.

| | | Predicted Class | | | | | | | | | |
|---|---|---|---|---|---|---|---|---|---|---|---|
| | Class | T | S(L[1]) | S(R[2]) | P(L) | P(R) | C(L) | C(R) | Total | PA | OE |
| **Actualm class** | **T** | 1219 | 0 | 2 | 0 | 8 | 0 | 0 | 1229 | 99.19% | 0.81% |
| | **S (L)** | 0 | 58 | 0 | 0 | 0 | 0 | 0 | 58 | 100% | 0% |
| | **S (R)** | 2 | 0 | 973 | 2 | 3 | 1 | 0 | 981 | 99.18% | 0.82% |
| | **P (L)** | 0 | 0 | 0 | 208 | 16 | 2 | 4 | 230 | 90.43% | 9.57% |
| | **P (R)** | 3 | 0 | 3 | 10 | 784 | 2 | 8 | 810 | 96.79% | 3.21% |
| | **C (L)** | 2 | 0 | 1 | 0 | 1 | 119 | 3 | 126 | 94.44% | 5.56% |
| | **C (R)** | 2 | 2 | 2 | 0 | 1 | 0 | 979 | 986 | 99.29% | 0.71% |
| | **Total** | 1228 | 60 | 981 | 220 | 813 | 124 | 994 | 4420 | | |
| | **UA** | 99.27% | 96.67% | 99.18% | 94.55% | 96.43% | 95.98% | 98.49% | | **OA** | 98.19% |
| | **CE** | 0.73% | 3.33% | 0.82% | 5.45% | 3.57% | 4.02% | 1.51% | | **KS** | 97.70% |

[1] Left-handed use. [2] Right-handed use.

### 3.3. Pedestrian Walking Speed Estimation Results and Analysis

Table 5 shows the performance of the proposed step detection algorithm for every combination of pose information. The twelve subjects taking part in this experiment and the ground truth and estimated step count are list in the Table 5. The average precision of texting, swinging, pocket, and calling modes were 99.78%, 99.32%, 99.85%, and 99.78%, respectively. The experimental results demonstrate that the performance of the proposed algorithm was not greatly affected by any pose and achieved a high level of precision (99.68%) consistently over any combination of poses. The result of the eighth subject with the swing pose was the worst (86.14%) because the swing step of this subject was homolateral. In this subject's walking pattern, the swing arm and leg are on the same side, but the subject regulated the swinging posture deliberately during the experiment.

**Table 5.** Metrics for evaluating the adaptive step detection method with different pose.

| Subject | Truth Steps (T) | Estimated Steps (T) | Truth Steps (S) | Estimated Steps (S) | Truth Steps (P) | Estimated Steps (P) | Truth Steps (C) | Estimated Steps (C) |
|---|---|---|---|---|---|---|---|---|
| S1 | 128 | 128 | 124 | 123 | 121 | 124 | 125 | 126 |
| S2 | 113 | 115 | 112 | 112 | 112 | 112 | 114 | 115 |
| S3 | 118 | 119 | 116 | 115 | 117 | 115 | 115 | 115 |
| S4 | 103 | 103 | 102 | 101 | 100 | 102 | 102 | 103 |
| S5 | 119 | 118 | 111 | 112 | 109 | 103 | 121 | 117 |
| S6 | 126 | 126 | 121 | 117 | 123 | 117 | 122 | 123 |
| S7 | 126 | 125 | 117 | 116 | 126 | 127 | 122 | 119 |
| S8 | 106 | 108 | 101 | 115 | 106 | 106 | 104 | 105 |
| S9 | 116 | 116 | 108 | 109 | 107 | 109 | 111 | 112 |
| S10 | 102 | 103 | 103 | 107 | 102 | 104 | 103 | 104 |
| S11 | 110 | 110 | 103 | 106 | 106 | 108 | 108 | 108 |
| S12 | 121 | 120 | 114 | 108 | 109 | 113 | 116 | 113 |
| Average | 115.67 | 115.92 | 111 | 111.75 | 111.5 | 111.67 | 113.58 | 113.33 |

In this test, participants carried the 360-degree prism and walked 80 m with four different poses. a comparison of the performance using five step length estimation models was carried out, and we analyzed their precision in texting, pocket, swinging, and calling modes. Figure 10 illustrates the maximum and minimum values, lower and upper quartiles, and median walking length obtained using the five step length models, including the static [46], Weinberg [28], Tian [47], Kim [29], and Chen [22] models, in four posture modes. In Figure 10, the red line represents the median of estimation distance and the green line represents the actual distance (80 m). The performance of the Chen method showed the highest precision and greatest robustness by significant margins. For texting mode, the length-estimating errors of the Chen method were within (76.75 m, 85.95 m) and the 25% and 75% errors were 78.46 m and 82.83 m. For calling mode, the length-estimating errors were within (75.36 m, 87.37 m) and the 25% and 75% errors were 77.50 m and 82.40 m. For pocket mode, the length-estimating errors were within (69.62 m, 87.82 m), and the 25% and 75% error were 77.05 m and 82.39 m, respectively. For swinging mode, the length-estimating errors were within (73.92 m, 109.51 m), and the 25% and 75% error were 76.97 m and 84.15 m, respectively.

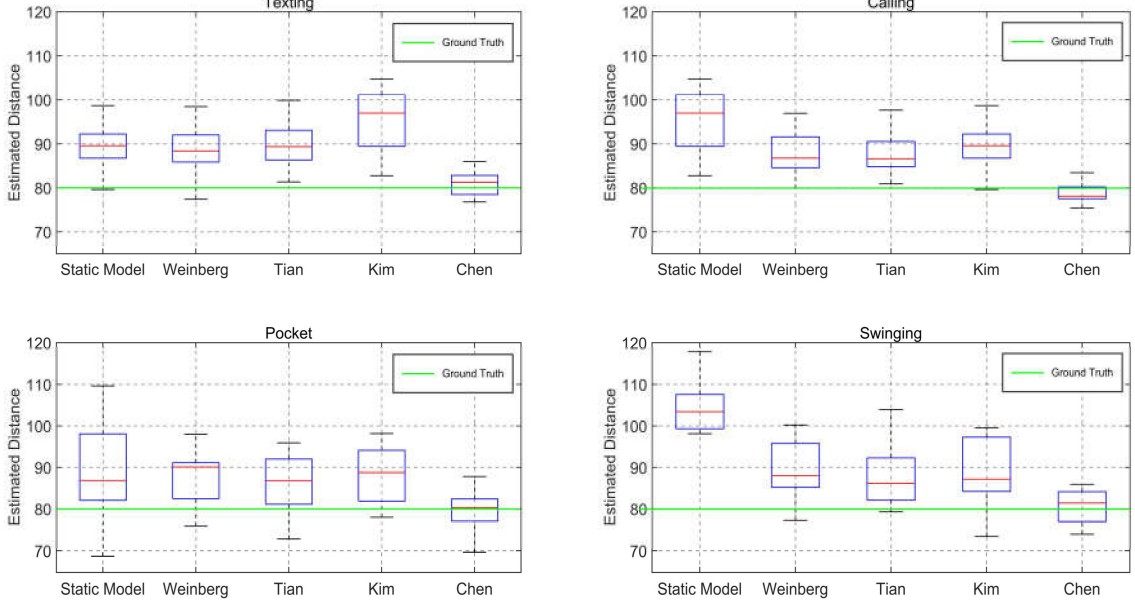

**Figure 10.** Comparison of the walking distance estimation method in different poses.

The definition of the evaluation parameters is listed in Table 6. The absolute error rate and the standard deviations (std) of four different poses are compared in Table 7. The average error column presents the mean absolute percentage error of four poses.

**Table 6.** Evaluation parameters definition.

| Evaluation Parameters | Definition |
|---|---|
| Estimated speed | $SL$ |
| Ground truth speed | $SL_{truth}$ |
| Absolute error rate | $SL = \frac{|SL - SL_{truth}|}{SL_{truth}}$ |
| Mean absolute percentage error | $M_{SL} = \frac{1}{N} \sum\limits_{k=1}^{N} E_k$ |
| Estimated speed std | $\sigma_{SL} = \sqrt{\frac{1}{N} \sum\limits_{k=1}^{N} \left(SL_k - \overline{SL}\right)^2}$ |

**Table 7.** Step length error comparison.

| Method | Texting | | Calling | | Pocket | | Swinging | | Average |
|---|---|---|---|---|---|---|---|---|---|
| | Error (%) | Std (m) | Error (%) | Std (m) | Error (%) | Std (m) | Error (%) | Std (m) | Error (%) |
| Static Model | 11.42 | 5.11 | 19.26 | 7.12 | 14.54 | 11.72 | 30.82 | 16.09 | 19.01 |
| Weinberg | 11.49 | 5.62 | 9.36 | 5.34 | 11.15 | 6.74 | 14.68 | 11.87 | 11.67 |
| Tian | 12.60 | 5.57 | 9.89 | 5.35 | 9.89 | 7.50 | 10.24 | 8.07 | 10.66 |
| Kim | 19.26 | 7.12 | 11.42 | 5.11 | 10.69 | 6.59 | 17.05 | 17.52 | 14.61 |
| Chen | 3.14 | 2.91 | 3.66 | 3.40 | 4.81 | 5.07 | 6.99 | 9.18 | 4.65 |

The results indicate that the absolute error of the Chen [22] method was much lower than those of the other four methods, which were 3.14%, 3.66%, 4.81%, and 6.99% in the four posture modes. The static model resulted in particularly high errors with an average of 19.01%, whereas the Kim method obtained an average error of 14.61%. The Tian and Weinberg approaches achieved better results, with average error rates of approximately 11%. The average error rate of the distance estimation was reduced to 4.65% with the Chen step length estimating algorithm.

Tables 8 and 9 show the proposed pedestrian walking speed estimation result from the second test. Twelve subjects took part in this experiment and the average of the ground truth and the estimated speed of four poses are listed in Table 8. The average PWS estimation absolute error was 0.061 m/s. The proposed algorithm combined with the texting pose produced the best result. The error of the texting pose was between 0.006 m/s and 0.09 m/s, and the 50% and 95% errors were 0.031 m/s and 0.089 m/s, respectively. The estimation errors of the calling mode were within (0.022 m/s, 0.122 m/s), and the 50% and 95% error were 0.039 m/s and 0.068 m/s, respectively. The estimation errors of the pocket pose were within (0.013 m/s, 0.176 m/s), and the 50% and 95% errors were 0.046 m/s and 0.126 m/s. The mean error of the swing motion was 0.094 m/s, which was higher than those of the other three posture modes, and the 50% and 95% errors were 0.058 m/s and 0.11 m/s, respectively. The means, standard deviations, and variance of the error are also listed in Table 9.

**Table 8.** Metrics for evaluating the pedestrian walking speed estimation method with different pose.

| Subject | Truth Speed (T) (m/s) | Estimated Speed (T) (m/s) | Truth Speed (S) (m/s) | Estimated Speed (S) (m/s) | Truth Speed (P) (m/s) | Estimated Speed (P) (m/s) | Truth Speed (C) (m/s) | Estimated Speed (C) (m/s) |
|---|---|---|---|---|---|---|---|---|
| S1 | 1.12 | 1.14 | 1.12 | 1.15 | 1.12 | 1.16 | 1.12 | 1.18 |
| S2 | 1.34 | 1.38 | 1.34 | 1.32 | 1.34 | 1.36 | 1.34 | 1.24 |
| S3 | 1.24 | 1.21 | 1.24 | 1.18 | 1.24 | 1.23 | 1.24 | 1.26 |
| S4 | 1.43 | 1.40 | 1.43 | 1.40 | 1.43 | 1.38 | 1.43 | 1.38 |
| S5 | 1.32 | 1.41 | 1.32 | 1.44 | 1.32 | 1.27 | 1.32 | 1.38 |
| S6 | 1.21 | 1.30 | 1.21 | 1.18 | 1.21 | 1.29 | 1.21 | 1.27 |
| S7 | 1.16 | 1.17 | 1.17 | 1.10 | 1.17 | 1.28 | 1.17 | 1.11 |
| S8 | 1.48 | 1.42 | 1.48 | 1.44 | 1.48 | 1.52 | 1.48 | 1.59 |
| S9 | 1.36 | 1.38 | 1.36 | 1.32 | 1.36 | 1.18 | 1.36 | 1.38 |
| S10 | 1.37 | 1.35 | 1.37 | 1.33 | 1.37 | 1.25 | 1.37 | 1.88 |
| S11 | 1.49 | 1.55 | 1.49 | 1.55 | 1.49 | 1.53 | 1.49 | 1.52 |
| S12 | 1.31 | 1.33 | 1.31 | 1.27 | 1.31 | 1.29 | 1.31 | 1.26 |

**Table 9.** Walking speed estimation error of the different pose.

| Stat. | Texting | Calling | Pocket | Swing | Average |
|---|---|---|---|---|---|
| Mean (m/s) | 0.042 | 0.048 | 0.063 | 0.094 | 0.061 |
| Std (m/s) | 0.028 | 0.028 | 0.051 | 0.133 | 0.074 |
| Var (m/s)$^2$ | 0.001 | 0.001 | 0.003 | 0.018 | 0.005 |
| Max (m/s) | 0.090 | 0.122 | 0.176 | 0.507 | 0.507 |
| 95th (m/s) | 0.089 | 0.068 | 0.126 | 0.110 | 0.121 |
| Median (m/s) | 0.031 | 0.039 | 0.046 | 0.058 | 0.043 |
| Min (m/s) | 0.006 | 0.022 | 0.013 | 0.015 | 0.006 |

## 4. Discussion

Overall, our studies established an adaptive pedestrian walking speed estimation system on a consumer-grade smartphone. Evaluations of our methods with different criteria showed that an adaptive step detection method coupled tightly with pose context can accurately estimate pedestrian walking speed. From the results, extracted features from multi-sensors, including accelerometer, gyroscope, light, and proximity sensors, express the four basic pose features accurately. We also demonstrated the adaptive step detection method aided with real-time pose context recognition, which was not greatly affected by any pose and achieved a high level of precision consistently over any combination of poses.

Numerous recent ML-based works [10–14] predicted the PWS with a pre-trained black-box model; however, automatic feature extraction, generalization, and unbiased dataset remained a challenge on this task. Therefore, we tackled the problems of multi-pose context pedestrian walking speed estimation in a model-based way [20–29]. Compared with previous studies [28,29], knowing pose context beforehand can improve the PWS estimation precision. The results confirm that the constraints on how the smartphone is carried were reduced in this task, and average absolute speed error achieved was 0.061 m/s. Multi-pose PWS estimation has great potential for indoors smartphone positioning and tracking systems. Our future study will focus on fusing the PWS with multiple measurements (e.g., absolute position, angle-of-arrival, ranging, and time-of-advent) for smartphone indoor positioning.

Although experiments have proven that our pose identification method performed well on four basic trained poses, as an ML application, the performance of our method may be limited for an untrained pose or activity. In future work, we will focus on improving the generalization of the system and extending our method to additional postures and activities. Besides, we also aim to set up an unbiased dataset that covers a wider range of ages, heights, genders, and handedness of the subjects.

## 5.Conclusions

This paper proposed an adaptive pedestrian walking speed estimation solution aided by pose awareness on the smartphone platform. In this solution, the real-time smartphone-posed context was coupled with an adaptive step detection method to precisely estimate the pedestrian walking speed using the multi-sensors tightly. Field tests were carried out to verify the proposed pose context awareness and adaptive step detection algorithms. The proposed awareness solution was reliable and could achieve a 98.85% overall accuracy and a 98.46% kappa statistic. The performance of the proposed adaptive step detection algorithm was almost unaffected by the pose in the test, and was able to consistently achieve a high level of precision (99.68%) over any combination of posture in the tests. The performance of the proposed solution developed on a commercial smartphone resulted in a mean absolute of 0.061 m/s over the different posture modes in real time. In future work, we will focus on improving the robustness of the system and extending it to additional postures and activities. Additionally, we plan to fuse the pedestrian walking speed with other measurements for indoor positioning.

**Author Contributions:** This paper is a collaborative work by all the authors. G.G. proposed the idea, implemented the system, performed the experiments, analyzed the data, and wrote the manuscript. R.C. and L.C. aided in proposing the idea, gave suggestions, and revised the rough draft. F.Y., M.L., Z.C., and Y.P. assisted with certain experiments.

**Funding:** This research was funded by the National Key Research and Development Program of China (grant nos. 2016YFB0502200 and 2016YFB0502201) and the NSFC (grant no. 91638203).

**Conflicts of Interest:** The authors declare that they have no conflict of interest to disclose.

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
