# Peer review of "A Pose Awareness Solution for Estimating Pedestrian Walking Speed"

_remotesensing, doi:10.3390/rs11010055_

Round 1
Reviewer 1 Report
This paper presents an approach for estimating smartphone pose and human walking speed, based on the analysis of measurements taken from sensors embedded in the smartphone itself.
The proposed method is based on the rationale that detecting smartphone pose (e.g. texting, in a pocket...) can be exploited to improve the walking speed estimation. To be more specific, knowledge about smartphone current use (e.g. texting...) can be used to modify the approach for computing the current speed.
Despite it can be possible to estimate smartphone movements by directly integrating inertial sensor measurements, this typically leads to drifting estimates, due to the significant noise on such measurements. This observation motivated the search in several works for methods determining walking speed by directly correlating such speed with sensor measurements via machine learning techniques. This work extends this kind of approaches by integrating also smartphone pose detection.
There are several works in the literature either trying to estimate human walking speed from inertial sensor measurements, or smartphone pose, however the kind of combination proposed here is probably quite new. The proposed idea is quite intuitive and the obtain results are relatively good.
A list of observations/suggestions is reported in the following:
- Description of modes starting at line 121: there are certain relevant cases that have not been listed, for instance: a) smartphone can be hand held along the body, close to the leg, even without swinging, b) women often carry their smartphone in a bag. I guess these cases can be considered similarly to those already listed, hence it would be good to add them to the list, if appropriate.
- line 141-142: in such conditions the device can be inside a bag as well.
- line 148, Figure 2. Differently from conventional reference systems, the reference frame defined in figure 2 is not right-handed. Despite not being a must, this is quite strange. Furthermore, this is also in contrast with the top-left plot in Figure 4: z axis in such figure is clearly the one approximately corresponding to the gravitational acceleration, whereas according to Figure 2 such axis should be the Y one.
- line 170, it is not clear wha you mean for "detached", please rephrase the sentence in order to make it more clear.
- line 191, equation (1). Despite at line 186 authors claim that they want to detect pedestrian walking state, they have not clearly stated their usage of equation (1). A reader can probably understand it by his/her own, however, for the sake of readability of the paper, I suggest to clearly specify in the text the usage of such condition.
- Despite such work is quite interesting, in my opinion it is quite risky to consider step detection criteria which are not invariant to certain rotations, indeed, even in "a similar pose" the smartphone orientation might be different for different persons. For instance, in the "swinging" case, one might held with any kind of pose with respect to the hand: probably certain poses are much more probable than others, but all the possible relative orientations between smartphone and hand are possible during swinging. Such relative orientation is probably approximately constant during swinging, but the choice of such specific orientation is personal, not the same for all the persons and not the same in all the cases. Consequently, I suggest either to specify that in the considered study a specific relative orientation between smartphone and hand have been assumed in the four considered cases (I also suggest to show a figure where an example of such relative positions are reported) or to extend the method to something independent on such relative orientation.
- line 286: if I am not wrong H in (14) is the height of the walking person: authors should specify if such height has been manually inserted in the algorithm for each user or estimated in some way by the algorithm itself.
- line 298: Despite a quite large number of persons have been involved in the experiment their age range is quite limited. If possible I suggest to extend the experiment to certain other older persons in order to be sure to have a more "unbiased" dataset.
- paragraph starting at line 338 confirms my personal opinion that detection criteria should be independent of certain rotations, for instance in order to make them robust with respect to left/right hand use. Alternatively, one might separately try to determine if it is being held by the left/right hand, however I am not so sure about the real advantages led by such estimate.
- Table 4 should be reported in just one page. The same observation applies to Table 5.
- Figure 9. If I am not wrong the authors have not provided the ground truth distances: please report them both in the text and in the figure.
- line 377 and Table 5: please specify the exact definition used for "error rate" and for "error" and "std".
- line 385: it is not clear to me if the authors are considering the either the average speed or its values all along the trajectories. Furthermore, in the second case I guess it should be related just to the average speed during each step... Please specify exactly what you are considering. Similarly to the previous observation, exact formulas for mean (average of the absolute value of the error?) and std in Table 6 should also be reported.
- line 406 (and maybe 23): claim about speed estimation error should be partially modified in order to clarify which kind of speed and how it has been computed.
- a minor English revision might be needed. For instance,
line 29: I guess it should be "Due TO the development..."
line 83: I suggest to change "possesses" with a more appropriate term
line 122: I suggest to change "violent" with a more appropriate term
line 149: I suggest to change "posture" with a more appropriate term
Author Response
Dear reviewers:
Thanks for your thoughtful comments and suggestions for our manuscript. We have revised the manuscript according to the reviewers’ comments. The detailed responses to the editor’s and reviewers’ comments are listed below point by point. The manuscipt is attached.
Yours Sincerely
Guangyi Guo

Reviewer 2 Report
This is a good attempt towards developing a system for estimating the pedestrian walking speed. However, the paper has significant flaws in terms of writing style and flow. Better clarity in needed at various places. Some of terms are used vaguely and need to be corrected. I have following recommendations to the authors:
1) The writing style is confusing at some places and needs to be improved. For example: lines 164-166.
2) Line 15: Whose health information? Please make it clear.
3) Line 15: Just a PWS is not enough for positioning. Additional information would be required for positioning.
4) line 87: The word should be "extracted".
5) Line 107 is not making any sense. This should be re-written for better understanding.
6) Line 118: Unclear statement. Please re-write.
7) Section 3.2: Feature extraction: The authors should inform the reader (probably in the introduction of this section) about the features they are referring to. Since the features are not defined explicitly, line 134 appears to be too vague.
8) Line 149: Unclear. Please re-write.
9) Table 1: Please replace "Acceleration/Gyroscope" with "Accelerometer/Gyroscope"
10) Lines 164-166 are unclear and need to be re-written.
11) The equation (1) is presented to the reader without any explanation/justification. In the current form, the flow of the paper is disturbed. Further, the parameters used in this equation are not described completely.
12) What are the units of alpha in equation 11 and how should the reader decide on the value of alpha to be used? A discussion on this should be provided.
13) In equation 11, the units of angular rate and acceleration are different. How can the authors simply add the two values? Justification for this equation needs to be presented.
14) Line 225: It depends on the orientation of the phone. It need not be z axis always.
15) Equation (6): What is the parameter g?
16) Line 231: What is the reason for choosing this specific filter and specific parameters (order and frequency)? A justification is needed. The authors should provide a citation for the butterworth filter.
17) The time axis in figure 5 appears incorrect. The data in figures 4 and 5 is similar but the time axis are very different. Please check this.
18) line 239: What do authors mean by calculation efficiency?
19) Line 251: Meaning of total significant signal magnitude is unclear. Please write clearly.
20) Algorithm 1 breaks the flow of the paper and is introduced without any explanation. This should be corrected.
21) What is the parameter Twin in algorithm 1?
22)Line 291: What do authors mean by pretraining?
23) Tables 2 and 3: The authors use percentages and absolute numbers interchangeably in the confusion matrices, which is very confusing. I would advise the authors to use absolute numbers instead of percentages.
24) Tables 2 and 3: The authors should include user's accuracy, producer's accuracy, overall accuracy and kappa statistics to discuss about the classification accuracy. The current discussion on classification accuracy is incomplete and insufficient.
25) What do the numbers in table 4 represent? This should be explained in the text.
26) Figure 9: What do the authors mean by estimation distance? This should be explained clearly.
27) Table 6: Please include the values of the walking speed in the table. This would provide better information to the reader.
28) Line 338-349: The authors use the mean precision to specify the classification performance. To my understanding, that is not the correct way to discuss about classification accuracy.
Author Response

(The authors gave the same response as above.)

Reviewer 3 Report
Authors discuss about the Pedestrian Walking Speed (PWS) and its usefulness. Hence, it can be used as a “body speedometer” to reveal health status information and positioning indoors. The paper proposes a pose awareness solution for estimating pedestrian walking speeds using the sensors built in smartphones. The smartphone usage pose is identified by using a machine learning approach based on data from multiple sensors.
This approach is tightly coupled with real-time pose identification and pedestrian walking information using an adaptive step detection strategy. The multi-sensor data is collected from sixty-one male and thirty-eight female subjects and labeled with the pose type, and these data are used to evaluate the extraction features and train the classifier.
This is indeed a very challenging topic and area and also authors have solidly proved their findings. In addition, the implementation steps and the process are detailed in a thorough way showing readers the potential of this work. The results are confirmed by some figures as well as tables and further analysis.
Regarding the introduction, everything is clear as authors discuss all the previous works implemented in the corresponding field. In addition, authors could add further topics for future studies – maybe in a new paragraph so as to further enrich their contribution.
My concerns are the following:
· The writing style is confusing at some places and needs to be improved.
· Some of terms are used vaguely and need to be corrected.
· The equation (1) has no explanation. The parameters used in this equation are not described completely (the same stands for other equations as well).
· In the current form, the flow of the paper is disturbed.
· What do authors mean by calculation efficiency?
· What do authors mean by estimation distance?
· Please write in a more understandable way the algorithm.
· Table 4 should be reported in just one page. The same observation applies to Table 5.
Author Response

(The authors gave the same response as above.)

Round 2
Reviewer 1 Report
The authors have revised the paper according to my suggestions. In my opinion the approach should be extended in order to take into account of the different possible choices in holding a smartphone, even when considering one of the types in Fig 2 (for instance 90 degrees rotations along the z axis). Nevertheless, the paper has surely been improved and, even if with the limitation expressed above, the presented approach is interesting.
Author Response
Dear editor and reviewers:
Thanks for your thoughtful comments and suggestions for our manuscript. Grammar modifications are marked in red in the manuscript.
Yours Sincerely
Guangyi Guo
Response to reviewer #1
Comments and Suggestions for Authors
The authors have revised the paper according to my suggestions. In my opinion the approach should be extended in order to take into account of the different possible choices in holding a smartphone, even when considering one of the types in Fig 2 (for instance 90 degrees rotations along the z axis). Nevertheless, the paper has surely been improved and, even if with the limitation expressed above, the presented approach is interesting.
Response: Thanks for your significant comments and point out our limitation, your valuable suggestion has a good guiding effect on our future work and will try our best to solve those problems.
In future work, we will focus on improving the generalization of the system and extending our method to additional postures and activities. Besides, we also aim to set up an unbiased dataset which covers a wider range of ages, heights, genders and handedness of the subjects.
Reviewer 3 Report
Authors have addressed all the issues I wrote. So I vote for acceptance.
Author Response
Dear editor and reviewers:
Thanks for your thoughtful comments and suggestions for our manuscript. Grammar modifications are marked in red in the manuscript.
Yours Sincerely
Guangyi Guo